# Consumer demand for healthy beverages in the hospitality industry: Examining willingness to pay a premium, and barriers to purchase

Rob Hallak[1]*, Ilke Onur[2], Craig Lee[3]

**1** University of South Australia, Adelaide, South Australia, Australia, **2** Flinders University, Bedford Park, South Australia, Australia, **3** University of Otago, Dunedin, New Zealand

☯ These authors contributed equally to this work.
* Rob.Hallak@unisa.edu.au

**Data Availability Statement:** All relevant data are within the paper and Supporting Information files.

**Funding:** This work was supported by research funds provided by Le Cordon Bleu, Australia; the

## Abstract

This study empirically examines consumer demand for healthy beverages within the hospitality industry. The research investigates sociodemographic and motivational factors that influence consumers' 'willingness to pay a premium' (WTPP) price for healthy beverages using survey data from 1021 consumers in Australia and New Zealand (NZ). Water and juice are rated as representing 'healthy' beverages sold by hospitality businesses. Under 2% of respondents consider sugar free drinks as being healthy. Consumers rate a 'healthy' beverage as having low/no sugar, natural/no additives, or containing vitamins and minerals. Less than 1% of respondents identify 'probiotics' or 'organic' as a healthy beverage. Censored Poisson finds consumers who frequently eat out or are younger have higher WTPP. Healthy eating goals increase WTPP, whereas food economizing goals decreases WTPP. Food hedonism goals reduces consumers' WTPP, and gender differences moderates this relationship. The findings present new insights on consumer behavior and healthy consumption in hospitality.

## Introduction

The restaurant industry has experienced strong annual revenue growth worldwide. In the United States (US), the industry recorded a compound annual growth in revenue (CAGR) rate of 4.4% from 2009 to 2017 [1]. In Europe, Germany, and Spain's restaurant industry sales value growth in 2018 (compared to 2013) were 7.7% and 7.4% respectively [2, 3]. Revenue growth among large developing countries was also profound, with China, India, and Nigeria reporting growth of 33%, 29%, and 11% from 2013–2018 [4–7]. In the United Kingdom (UK), 20% of people ate out at least weekly, while in the US, more household food expenditure goes to eating away from home than at home [8, 9]. Dining away from home accounts for 27% of weekly household food and drink expenditure in Australia, amounting to over $45 billion per year [10–12]. However, increasing consumption of foods and beverages purchased from food-service establishments has been criticized for its negative impact on public health as meals eaten away from home are nutritionally poorer, larger in portion size, and higher in fat, sugar,

School of Management, University of South Australia; and Organic and Raw Trading Company Pty Ltd. The funds were awarded to RH and CL. The funders had no role in study design, data collection and analysis, decision to publish, or preparation of the manuscript.

**Competing interests:** The authors have declared that no competing interests exist.

and salt [13–15]. Furthermore, beverage options sold through hospitality and foodservice establishments tend to be higher in sugar and calories and are associated with increased risk of Type 2 diabetes and cardiovascular disease [16–18].

In recognition of the growing concerns around public health, the types of food and beverage products sold through the hospitality sector are experiencing major changes as consumer demand and public/political pressure is driving businesses to replace high calorie items with healthier alternatives [19, 20]. Although foods have gradually improved in terms of their healthiness, the availability of high calorie, sugar-sweetened beverages (SSB) remains prevalent in the hospitality industry [21].

Hospitality firms have opportunities to offer more functional foods, such as beverages containing low sugar and added vitamins and probiotics, as a useful strategy to attract the 'health dollar' [22]. Moreover, increasing the variety of healthy menu options, as well as providing more information about the healthiness of the products, is important in nudging consumers to choose healthily [23, 24]. A recent study by Hallak et al. [25] examined the supply of healthy beverages from a hospitality firm's perspective. Their telephone interviews with managers/ owners of 400 hospitality businesses discovered that the decision to sell healthy beverages is influenced by: 1) shelf life of the products, 2) if it is locally produced, 3) product profit margins, and 4) levels of consumer demand. Hallak et al. [25] also identified the need to further examine healthy beverage demand from the perspective of hospitality consumers.

Beverages account for over 40% of hospitality business revenues, and 'healthy' beverage products are among the fastest growing category of beverages [26]. They also potentially represent the easiest healthy product to incorporate as they are convenient to distribute, store, and sell, and it is relatively easy to change their presentation to meet customer demand [27]. These beverages include ready to drink teas, superfruit 100% juices such pomegranate juice, cherry juice, cloudy pear juice, bottled water, herbal teas, kombucha, and other products [28].

Hospitality consumers' decisions to purchase foods or drinks are influenced by attitudes and perceptions, taste and preferences, price, and their willingness to pay [29]. Despite evidence relating to consumer demand for healthier beverages, our understanding of consumer willingness to pay a premium for comparatively health beverage products remains uncertain. Are consumers willing to pay a higher price for a product that is purported as being healthy? If so, then at what price points will they select the healthier option, or, to opt for the less expensive 'unhealthy' alternative? While the healthiness of beverage products can be asssessed in terms of its nutritional profile, consumers' peceptions of what is considered as 'healthy' are highly subjective. This triggers the overarching questions of what beverages do consumers consider to be healthy, what criteria do they use to assesses healthiness, and will consumers be willing to pay a higher price? Specifically, this study will explore the following research questions:

**RQ1**: Which beverages that are sold in hospitality businesses do consumers consider to be 'healthy'?

**RQ2**: What criteria do consumers use to determine whether a beverage is considered to be 'healthy'?

**RQ3**: What are the obstacles faced by consumers when buying healthy beverages in hospitality businesses?

**RQ4**: What are the motivational, demographic, and behavioral variables that influence consumers' willingness to pay a price premium for healthy (vs unhealthy) beverages?

The context for this study is the fast-growing hospitality and foodservices sector. Businesses in this sector play an important role in providing consumers with healthy product options as

well as health information to enable informed consumer decisions [30]. The sector has also expanded due to the adoption of online food delivery platforms (including Uber Eats, Menulog, and Deliveroo), which experienced a 72% revenue increase between 2014 and 2019 [31]. Thus, the accessibility and availability of foodservice menu items has led to increasing consumption of foods away from home, and this is expected to continue to expand post COVID-19 [32].

Data for this study were collected in 2019 through an e-survey of 1021 consumers in Australia and New Zealand. Results were analyzed through SPSS and Stata using descriptive statistics, factor analysis, and censored Poisson regression to determine the effects on consumers' 'willingness to pay a premium' (WTPP) for healthy beverages. The study presents new knowledge on consumer attitudes and behaviors toward purchase of menu items from hospitality/foodservice establishment. It presents new insights on 'healthiness' of beverages from a consumer perspective, and in understanding what consumers consider to be 'healthy'. The research also explores the price premium at which consumers will select healthy compared to unhealthy options, as well as the perceived barriers to purchasing. These insights are important for both beverage manufacturers, hospitality firms, as well as public health authorities aiming to support the supply and demand of healthier products and in nudging consumers to make healthier choices.

## Literature review

### Consumer demand for healthy beverages

Consumer demand for healthy beverages such as ready to drink teas, super-fruit 100% juices, kombucha and other products have experienced significant growth in recent years [28]. Growth in demand has witnessed entrepreneurs and innovators entering the market, with the number of new suppliers in the last five years increasing at an annualized rate of 1.8% [33]. In addition, large soft drink manufacturers have launched new product lines of healthier products and have diversified through acquisitions of niche healthy beverage manufacturers. For example, in 2018 Coca-Cola Australia expanded into the kombucha market through their acquisition of a South Australian based kombucha manufacturer [34]. This followed PepsiCo in the US purchasing KeVita, a California based leading kombucha manufacturer in 2016 [35].

Despite the growth in consumer demand and expansion of beverage products, research on consumers' WTPP for healthy beverages and the factors that influence their purchase behaviors in a hospitality and foodservice context remains limited. Studies examining consumers' WTPP for healthy food tend to focus on a household shopping context [36–38] and general food intake, without distinguishing between food and beverages, or excluding beverages entirely [39–42]. Thus, it is relatively unknown how consumers will respond to the hospitality sector's strategies to attract the 'health dollar', i.e., businesses marketing and selling beverages purported as being healthy (e.g., containing low sugar, added vitamins and probiotics etc.). Consumers' purchase behaviors in hospitality presents a unique challenge in promoting healthy beverages, especially when dining out is associated with pleasure and enjoying an experience as opposed to healthy eating, or, when research in this area uses generic terms such as 'healthy food' and 'healthy menus' without distinguishing between food and beverage items. This distinction is important when beverages can account for over 40% of hospitality business revenues [26].

Despite this knowledge gap, our review of the literature found factors influencing consumer behavior towards healthy food and beverages can be broadly organized into 1) sociodemographic factors (age, gender, education, income) and 2) psychological factors (including beliefs, attitudes toward health and wellbeing) [36, 43–50]. There is also limited research on

the barriers to purchasing healthier beverages from the perspective of the hospitality consumer. The review of the literature and proposed hypotheses are presented in the following sections.

## Sociodemographic factors influencing healthy product purchases

Research on how sociodemographic variables influence consumers' purchases of healthy food and beverages reveals mixed results. Age and gender have been linked to consumer attitudes towards purchasing healthier food and drink options, although in different ways [43]. Some evidence suggests women are more inclined to make healthy food and drink choices, have a much higher preference for healthy products, and are more likely to purchase these regularly [36, 37, 44]. Women also report more positive attitudes and a greater willingness to purchase healthy food and beverages [45, 46]. However, while some studies have identified that women are less inclined to pay a premium for organic wine [51], other studies found that gender plays no role in determining consumers' WTPP for healthy/organic food products [38, 39]. Thus, the different contexts in which studies are conducted (i.e., food, beverages, wine) highlight the nuances in gender differences and the need to further investigate.

Some studies suggest that older consumers report a higher interest and acceptance of healthier food and beverage products, especially those that have purported disease risk reduction properties [47, 48]. Older consumers are also more likely to choose healthy options over non-healthy alternatives [49]. This may be explained by two main reasons: 1) Some older adults have a higher awareness and regard for dietary health; and 2) Older adults may have experienced greater exposure to healthy and functional food products as compared to younger consumers [44]. Studies also found that younger adults are less inclined to pay a premium for healthier food options [36, 51].

In contrast to gender or age, the effects of education and socioeconomic status on consumer attitudes towards healthy drinks is less established. Some studies report a positive relationship between levels of education and willingness to purchase nutrient enhanced foods and drinks [52–54], while others find no significant associations [39, 51]. Income is found to be positively associated with acceptance of functional food and drinks in a small number of studies [53, 55].

The literature discussed in this section forms the basis for hypothesizing the relationships among socio-demographic factors and consumers' WTPP for healthy beverages in a hospitality setting. Women may be more inclined to purchase healthy beverages, especially if these are linked to low sugar, intestinal well-being, weight loss and bone health [44], which is a common marketing strategy for healthy beverages served in foodservice settings [22]. Older adults may be more willing to pay a premium for healthy beverages perceived as functional (e.g., improve biological functions or reduce disease risk) compared to younger adults who have a low propensity to consume healthy foods [26]. Consumers on higher incomes may have higher propensities to purchase healthy beverages when dining out, especially since dining in hospitality and foodservice establishments is a discretionary income activity. Thus, the review of extant literature provides the basis for the following hypotheses to be examined in the context of healthy beverages purchased in the hospitality sector.

$H_1$: *Women will have a significantly higher willingness to pay a price premium for healthy beverage products compared to men*

$H_2$: *Older consumers will have a significantly higher willingness to pay a price premium for healthy beverage products as compared to younger consumers*

$H_3$: *Consumers' level of income will have a positive relationship with their willingness to pay a price premium for healthy beverage products*

## Psychological factors influencing healthy product choices

In addition to demographic factors such as age, gender, and income, research suggests that a desire to 'feel good' about oneself can be a powerful motivator in making healthy dietary choices [56, 57]. Consumer attitudes toward health and wellbeing has a powerful effect on purchasing behavior intentions for a wide range of food categories [23]. For example, an individual's 'health value'—defined as the extent to which the consumer places a premium on health—is a strong predictor of purchase behaviors, particularly in the context of meals away from home [24, 41]. Indeed, striving to pursue a healthy lifestyle for personal health reasons is an important motivation for making dietary choices [57]. Food purchasing behaviors of health-oriented consumers includes opting for low fat foods, or foods with added vitamins or minerals [58, 42]. Interestingly, health-oriented consumers, such as those who exercise regularly, tend to be much more willing to purchase healthy beverage options, even at the expense of taste [58]. This contrasts with consumers who are motivated not by health goals, but by hedonism; this market segment will have greater interest in the taste and enjoyment of food and beverage [59].

The contradiction between healthy eating goals and hedonism (i.e., food for taste and enjoyment) presents interesting questions regarding the relationship between psychological factors and consumers' WTPP for healthy beverages in a hospitality setting. It is intuitive to assume that consumers motivated to follow a healthy lifestyle or healthy eating are interested in purchasing healthy beverages [58]. However, does this hold true in various purchasing contexts, such as in a hospitality dining setting? Will hospitality consumers be willing to purchase healthy beverages at a premium price, and if this requires compromising taste and enjoyment?

Recent studies have shown that hospitality businesses may be reluctant to sell healthy beverages due to uncertainty around consumer demand (in addition to limited fridge space and profit margins) [25]. On one hand, if health-conscious consumers are in fact willing to pay a premium for healthy beverages, businesses could be missing out on a potential untapped revenue stream from high yield consumers. On the other hand, if dining in a hospitality setting is perceived as an opportunity to indulge, overriding healthy eating goals, then the business case for supplying higher priced healthy beverages to consumers may be financially unsound.

To explore these tensions, we propose the following hypotheses:

$H_4$: *Consumers' motivations toward healthy eating will have a significant positive effect on willingness to pay a price premium for healthy beverages*

$H_5$: *Consumers' motivations toward food hedonism will have a significant negative effect on willingness to pay a price premium for healthy beverages.*

## Barriers affecting healthy food choices

In addition to motivations around health and wellbeing, as well as hedonism and enjoyment, consumers may be motivated by the price of a particular product. Healthy beverages are perceived to be more expensive than unhealthy alternatives [58], with evidence suggesting affordability remains a key factor in consumer consumption of sugar sweetened beverages as opposed to healthy alternatives [60]. The relatively low price of unhealthy beverage choices, and the relative higher prices of healthy alternatives, appears to be a consistent factor in shaping consumer attitudes and purchasing behaviors towards healthy drinks. Thus, consumers motivated by price in making their purchase decisions for food and beverages would be less inclined to pay a premium. This is referred to as consumers' 'economizing goals' [59]. Hospitality and foodservice settings generally follow this trend, in that unhealthy beverage choices

(such as sugar sweetened beverages) are usually the lowest priced beverage items on the menu. This forms the basis for the following hypothesis:

$H_6$: *Consumers' motivations toward food economizing will have a significant negative effect on willingness to pay a price premium for healthy beverages*

## Methods

### Sample and instrument

Data for this study were collected through an electronic survey of consumers in Australia and NZ. The two countries were chosen due to the significant growth in the hospitality sector and growing demand for healthy food and beverage products [28, 61]. The researchers contracted the services of Qualtrics Market Research who distributed the electronic survey to consumer panels from Australia and New Zealand, representative by age, gender, and region. Participants were required to provide consent before proceeding with the survey. In accordance with the objectives and scope of the study, the sample frame was limited to hospitality and food service consumers. Thus, a screening criterion was used–"How often would you eat out in cafes/restaurants or purchase take-away foods?"- to ensure only active consumers participated in the research. Results from 1021 survey responses were used for the analysis.

A questionnaire was developed based on common themes from hospitality management and food research, following research designs adopting psychological frameworks for attitudes and behaviors toward healthy products [25, 41, 59]. The term 'healthy' was not defined or made explicit to respondents in order "to learn without the complications of definitions what current diners wanted and what their barriers were" [62]. Instead, the questionnaire asked respondents to self-report "What drinks/beverages from a restaurant/cafe menu do you consider as being 'healthy'?" and "Why do you consider these drinks as 'healthy'?". Using an open-ended question for healthy beverages was a deliberate approach to understand the concept of 'healthy' from a consumer perspective. The questionnaire captured information on consumers' frequency of purchases, typical spend, and the types of beverage products they purchase. Information on the respondents' income, gender, and age were also captured as these demographic variables are expected to influence consumers' willingness to pay, as specified in the research hypotheses. A copy of the research instrument used for data collection is included in S3 Appendix. Ethics compliance for this study was assessed and approved by the University of South Australia Business School Ethics Committee (REF 039/20019), and follows the guidelines stipulated by the Australian Government's National Statement on Ethical Conduct in Human Research 2007 (updated 2018). The study is also in compliance with Health Research Council of New Zealand Research Ethics Guidelines.

### Predictor variables

In addition to demographic and behavioral variables, the research instrument captured consumers' psychological variables relating to attitudes and goals toward health and nutrition, as these goals are hypothesized to influence willingness to pay. These include validated scales from the consumer psychology literature that capture 'healthy eating goals', 'food hedonism goals' and 'food economizing goals' [59]. *Food Hedonism Goals* were measured through two items representing the level of an individual's hedonistic approach to food and drink consumption. These measures include "I eat what I like and do not worry about the healthiness of food", and "The healthiness of a food has little impact on my food choices" (1 = Strongly disagree, 7 = Strongly Agree) [63, 59]. This variable was reverse coded for the regression analysis.

*Food Economizing Goals* were measured through two items and captures the extent to which consumers are motivated by price as opposed to health when making purchase decisions. These include "It's important to me that the food I eat on a typical day is not expensive" and "It's important to me that the food I eat on a typical day is cheap" (1 = Strongly disagree, 7 = Strongly Agree) [64, 59].

*Healthy Eating Goals* captures individuals' motivations to choose healthy food and drink products and maintain a healthy lifestyle. This was measured by three items such as "It's important to me that the food I eat on a typical day contains vitamins and minerals", "It's important to me that the food I eat on a typical day is good for my appearance", and "It's important to me that the food I eat on a typical day is nutritious" (1 = Strongly disagree, 7 = Strongly Agree) [64, 59].

### Criterion variable—willingness to pay a premium (WTPP) index

In this paper, we implement a novel contingent valuation (CV) method to estimate the value consumers place on healthy beverages. CV is a stated preference technique commonly used when revealed preference data is not readily available or hard (sometimes impossible) to obtain [65–67]. In other words, the goal of our contingent valuation is to measure the compensating variation for healthy beverages compared to unhealthy ones. Respondents were provided with a set of questions relating to their willingness to purchase a beverage at various price points: "*Suppose that the price of an 'unhealthy' drink (i.e., high in sugar, artificial ingredients) from a café/restaurant/takeaway is $5. Now, consider an alternative drink that was healthier (i.e., low in sugar, organic, has proven health benefits etc.), what price would you be willing to pay for the healthy drink?*". Thus, respondents were given a reference point for an unhealthy beverage of $5, and then asked to indicate their willingness to purchase (never/ occasionally/ most of the time) a comparatively healthy beverage for each incremental price point ($5, $6... $10). The reference point of $5 was selected as it is the average price of a typical sugar sweetened beverage sold in hospitality venues in Australia and New Zealand. The overall prices of food and beverages in the two countries are on par, in addition, the same scale was used for both countries as the AUD and NZD currencies are of approximately equal value (1 AUD = 1.04 NZD). To create a dichotomous WTPP variable, for each price point we have identified a respondent as willing to spend the specific amount if his/her response was 'occasionally' or 'most of the time'. Similarly, if the response was 'never', we have categorized it as the respondent is not willing to pay that specific amount for the healthy beverage.

The upper limit of the price scale was $10, equivalent to double the price of the 'unhealthy' beverage. As the response for this question ranges from $5 (same price as an unhealthy beverage) to $10 (double the price), we have identified the highest amount each respondent is willing to spend for the healthy beverage, and then subtracted $5 (the price of the unhealthy beverage) from this amount to identify the healthy beverage premium. For the lowest figure, $5, there is a group of participants who responded saying they would never buy the healthy beverage. In other words, these individuals would be unwilling to purchase healthy beverages even if the price is equivalent to the unhealthy option. At the same time, we have respondents who would buy it for the equal price of $5. To differentiate between these two groups of respondents, we added one (1) to all respondents willing to buy the healthy beverage at $5 or more. As a result, our dependent variable, WTPP, is indexed and ranges from 0 to 6 (Table 1).

Table 1 shows that the lowest price point of our scale ($5) is coded as '0' and represents those respondents who are not willing to buy the healthy beverage at all, not even if its price was the same as the unhealthy drink ($5). The indexed measure ranges from 1–6 where '1' represents those who are willing to buy the healthy drink if the price is $5, through to '6' where

consumers willing to pay $10. The 'N' for each price point is represented by the number of respondents who indicated that this price is the maximum that they are willing to pay.

## Control variables

We introduce various control variables for the regression analysis including age, country (NZ or AU), education, gender, employment status, income, marital status, having children, spending when dining out, and the frequency of eating out. For the empirical analysis, we include these variables in a certain sequence to observe any effects on our regression models. Controlling for variances in respondents' sociodemographics is necessary as the literature suggests nuances in consumer behavior toward purchases of health products [see 36, 45, 46]. In addition, the dependent variable (WTPP) may be influenced by the consumers' existing behaviors regarding 1) how frequently they purchase from hospitality and venues; and 2) The average amount spent during purchase. Thus, frequency of purchase and average spend were treated as controls.

## Model description and analysis techniques

Data were analyzed through SPSS and STATA. Descriptive analysis, factor analysis, and censored Poisson regression were used. Our dependent variable (WTPP), as shown in Table 1, is composed of nonnegative integers ranging from 0 to 6. Given the count data nature of the criterion variable, we use the Poisson regression method. The distribution of count data is discrete (not continuous), and it is limited to nonnegative values. Thus, the main assumption of normal distribution in a linear regression analysis (such as ordinary least squares–OLS) is likely to be violated. The OLS would also produce negative predicted values, which are impossible. For these reasons, the use of a nonlinear estimation model of Poisson is preferred [68, 69].

In addition, considering that the criterion variable is top censored (i.e., at $10), respondents willing to pay $10 may also be willing to pay a higher amount (e.g., $11, $12, $13, etc.). Since their willingness to pay an amount higher that $10 is not captured in our survey design, we analyze our data using censored regression to take the top coding into consideration [69]. Below, we formally present our censored Poisson regression analysis.

For the Poisson model, the probability mass function is the Poisson distribution:

$$\Pr(Y = y) = \frac{e^{-\mu}\mu^y}{y!}, \quad y = 0, 1, 2, \ldots$$

With the following exponential mean parameterization:

$$\mu_i = exp(x_i'\beta), \quad i = 1, 2, 3, \ldots, N$$

**Table 1. Willingness to pay a premium (WTPP) index.**

| Price point | Code | Description | N | % |
|---|---|---|---|---|
| $5 | 0 | Would not buy a healthy beverage even if same price as unhealthy beverage | 82 | 8.03 |
| $5 | 1 | Willing to purchase at same price of unhealthy beverage ($5) | 144 | 14.10 |
| $6 | 2 | Willing to purchase at $6, compared to unhealthy beverage of $5 *(premium = 20%)* | 143 | 14.01 |
| $7 | 3 | Willing to purchase at $7, compared to unhealthy beverage of $5 *(premium = 40%)* | 162 | 15.87 |
| $8 | 4 | Willing to purchase at $8, compared to unhealthy beverage of $5 *(premium = 60%)* | 143 | 14.01 |
| $9 | 5 | Willing to purchase at $9, compared to unhealthy beverage of $5 *(premium = 80%)* | 65 | 6.37 |
| $10 | 6 | Willing to purchase at $10, compared to unhealthy beverage of $5 *(premium = 100%)* | 282 | 27.62 |
| TOTAL | | | 1021 | 100% |

Thus, we end up with the following probability function:

$$f(y_i; \mu_i) = \frac{e^{-\mu_i}\mu_i^{y_i}}{y_i!}, \quad i = 1, 2, \ldots, n$$

Where $\mu_i = exp(x_i\beta)$, $x_i$ is a vector of exogenous variables and $\beta$ is a vector of unknown parameters. These unknown parameters are the regression coefficients that are estimated from a dataset using the method of maximum likelihood.

Since we do not observe all $y_i$ exactly, we introduce a censored Poisson model. To this end, assume the true $y_i^*$ is observed only below a censoring point $c_i$. Thus, we have the following:

$$y_i = \begin{cases} y_i^*, & if \ y_i^* < c_i \\ c_i, & if \ y_i^* \geq c_i \end{cases}$$

In our case, the data are censored from above at the value of 6 because of top-coding. In other words, for us $c_i = 6$. Ignoring the censoring would lead to inconsistent parameter estimates.

If $y_i$ is censored, we know that:

$$Pr(y_i \geq c_i) = \sum_{j=c_i}^{\infty} Pr(y_i = j) = \sum_{j=c_i}^{\infty} f(j) = 1 - \sum_{j=0}^{c_i-1} f(j) = 1 - F(c_i - 1)$$

To write down the maximum likelihood estimator (MLE), it is convenient to introduce an indicator variable $d_i$ such that:

$$d_i = \begin{cases} 1, & if \ y_i^* \geq c_i \\ 0, & otherwise \end{cases}$$

Then, the log likelihood function can be written as:

$$\mathcal{L}(\beta) = \sum_{i=1}^{n} [(1 - d_i)log f(y_i) + d_i log\{1 - F(c_i - 1)\}]$$

The estimation is by maximum likelihood and the initial values are taken from the Poisson model.

## Results

### Descriptive analysis

Data were collected from 1021 consumers across Australia (N = 808, 79.14%) and NZ (N = 213, 20.86%) (S1 Appendix). There was an even spread of male and female respondents, and a relatively even spread of respondents across the six age categories measured (ranging from 18 to 65 and over). Most respondents were in full- or part time work and the largest income group was those earning less than $40K in annual household income (25.27%), followed by households earning between $40-$60K annually (24.19%). Education wise, the largest group were those with a Certificate/Diploma (32.21%), followed by those who completed high school (26.84%), and those with a Bachelor degree (22.33%). To assess the representativeness of our sample, we compared the distribution of our Australia and NZ samples' gender and age variables to their respective country's census data using chi square difference tests. These tests produced non-significant results (S2 Appendix), indicating that the gender and age

distributions of our sample to be similar to the general population of Australia and NZ, supporting the representativeness of our sample.

In terms of purchasing behaviors, over 66% of respondents purchased from hospitality businesses at least once a week, with 20% of the sample purchasing 3 or more times per week. Almost 79% of respondents spent around $35 or less in each purchase. These results are consistent with Australian data indicating meals away from home counting for 27% of weekly household expenditure on food and beverages [10, 11].

## Which drinks are considered as 'healthy' and why?

Respondents were asked an open-ended question about which drinks sold at hospitality businesses they rate as being healthy, and the reasons why they rate them as being healthy. Tables 2 and 3 below categorize the responses received. As this was an open-ended question, respondents were free to give multiple answers; therefore, the percentages within the two tables represent the percentage of responses received for a particular drink category across all received responses. A high number of respondents identified water, including bottled water and sparkling water, as the main healthy beverage. Fruit juices, especially freshly squeezed, were also frequently reported (39%). Tea was more frequently reported than coffee, and some respondents identified wine and beer to be healthy (2.7% and 1.8%). Interestingly, only 1.7% of respondents reported sugar-free soft drinks as a healthy beverage.

When asked to state the reasons for listing a particular drink as healthy, a large proportion of the responses were because the drinks contain low or no sugar (47%) or the drink was natural or had no additives (42%). Very few respondents indicated that they considered a drink healthy because it contains probiotics (0.64%) or because it was 'organic' (0.13%).

## Predictors of willingness to pay a premium (WTPP) for healthy beverages

The review of the literature identified demographic and motivational variables as predictors of consumers' purchase behavior toward healthy food and beverage products. Motivational factors include Healthy Eating Goals (3 items), Food Economizing Goals (2 items), and Food Hedonism Goals (2 items). Principal Component Analysis with oblique (Oblimin) rotation was conducted on the seven observed variables. The results of the PCA supports a three-factor solution (eigenvalues > 1) as the best fit for the data and explains 78.9% of the variance in the model. The component matrix shows each observed item loads well onto its corresponding factor. The Cronbach's alpha for each factor is >.7, supporting scale reliability (internal consistency) (Table 4).

**Table 2. Types of drinks consumers consider healthy.**

| Drinks considered healthy | N | % |
|---|---|---|
| Fruit juice | 395 | 38.96 |
| Water | 555 | 54.36% |
| Tea | 158 | 15.48% |
| Smoothies | 86 | 8.42% |
| Coffee | 78 | 7.64% |
| Wine | 28 | 2.74% |
| Beer | 25 | 1.76% |
| Sugar-free soft drinks | 17 | 1.67% |
| Probiotics/kombucha | 31 | 3.04% |

**Table 3. Reasons consumers consider drink(s) as healthy.**

| Reason for drink(s) being healthy | N | % |
|---|---|---|
| Low or no sugar | 364 | 46.91 |
| Natural / No additives | 324 | 41.75 |
| Contains vitamins and minerals | 82 | 10.57 |
| Contains probiotics | 5 | 0.64 |
| Organically produced | 1 | 0.13 |

## Censored Poisson regression

Censored Poisson regression was used to examine the effects of the explanatory (and control) variables on the willingness to pay a premium (WTPP) dependent variable. The explanatory variables include the factor scores for Healthy Eating Goals, Food Economizing Goals, and Food Hedonism Goals. Results from Model 1 indicate that Healthy Eating Goals has a positive and significant effect on WTPP. In contrast, Food Economizing Goals and Food Hedonism Goals have significant negative effects on WTPP (Table 5).

In Table 5, we present the results of the censored Poisson regression. Model 1 examines the effects of the motivational goals (healthy eating, food economizing, and food hedonism) on WTPP. In Model 2, 3, and 4 we incrementally introduce control variables. The control variables are age, country (NZ or AU), education, gender, employment status, income, marital status, having children, spending when dining out, and the frequency of eating out. In Model 5, we interact our gender variable with the motivational factor variables to examine any moderation effects on WTPP. The female dichotomous dummy variable is equal to one if the respondent is a female, and zero otherwise.

For all Models, the coefficients for the healthy eating goals are consistently positive and significant with a similar magnitude throughout. Likewise, the coefficients for the food economizing variable are consistent throughout the five specifications, being significant and negative (as hypothesized). However, for the food hedonism variable we observe no statistical significance in models three and four, which is due to the addition of demographic variables, especially the age categorical variables. This suggests a closer association between age and the food hedonism variable compared to food economizing and healthy eating variables. When statistically significant, we observe the effect of food hedonism goals to be negative. Overall, the effect of hedonism on WTPP is not as robust as the effects we observe for the healthy eating

**Table 4. Factor analysis results.**

| Variable | Item Description | Mean (SD) | Factor 1 | Factor 2 | Factor 3 |
|---|---|---|---|---|---|
| **Healthy Eating Goals 1** | It's important to me that the food I eat on a typical day contains vitamins and minerals' | 4.7 (1.539) | .858 | | |
| **Healthy Eating Goals 2** | It's important to me that the food I eat on a typical day is good for my appearance | 4.24 (1.587) | .757 | | |
| **Healthy Eating Goals 3** | It's important to me that the food I eat on a typical day is nutritious | 5.08 (1.399) | .825 | | |
| **Food Economizing Goal 1** | It's important to me that the food I eat on a typical day is not expensive | 4.92 (1.369) | | .751 | |
| **Food Economizing Goal 2** | It's important to me that the food I eat on a typical day is cheap | 4.41(1.527) | | .788 | |
| **Food Hedonism Goal 1** | I eat what I like and do not worry about the healthiness of food | 4.19(1.668) | | | .461 |
| **Food Hedonism Goal 2** | The healthiness of a food has little impact on my food choices | 4.22(1.642) | | | .519 |
| *Eigenvalue* | | | 2.505 | 1.971 | 1.054 |
| *Total variance Explained* | | | 35.789 | 28.153 | 15.052 |
| *Cronbach Alpha* | | | .833 | .753 | .769 |

Note: Factor1 = Healthy Eating Goals, Factor 2 = Food Economizing Goals. Factor 3 = Food Hedonism Goals. Total variance explained = 78.994%.

**Table 5. Censored Poisson regression results for the determining factors of willingness to pay a premium (WTPP).**

| Variable | Model 1 | Model 2 | Model 3 | Model 4 | Model 5 |
|---|---|---|---|---|---|
| Healthy Eating Goals | **0.178**\*\*\* | **0.170**\*\*\* | **0.181**\*\*\* | **0.181**\*\*\* | **0.205**\*\*\* |
| | (0.0186) [19.48%] | (0.0186) [18.53%] | (0.0189) [19.84%] | (0.0207) [19.84%] | (0.0281) [22.75%] |
| Food Economizing Goals | **-0.0774**\*\*\* | **-0.0670**\*\*\* | **-0.0984**\*\*\* | **-0.103**\*\*\* | **-0.0765**\*\*\* |
| | (0.0174) [-7.45%] | (0.0175) [-6.48%] | (0.0180) [-9.37%] | (0.0191) [-9.79%] | (0.0282) [-7.36%] |
| Food Hedonism Goals | **-0.0439**\*\* | **-0.0382**\*\* | 0.00225 | -0.00374 | **-0.0637**\*\* |
| | (0.0176) [-4.30%] | (0.0177) [-3.75%] | (0.0183) | (0.0194) | (0.0276) [-6.17%] |
| *Spending when eating out* (Base: $20 or less) | | | | | |
| • $21 –$35 | | **0.167**\*\*\* | **0.145**\*\*\* | **0.117**\*\*\* | **0.110**\*\* |
| | | (0.0398) [18.18%] | (0.0404) [15.60%] | (0.0434) [12.41%] | (0.0435) [11.63%] |
| • $36 or more | | **0.0865**\* | 0.0743 | 0.0368 | 0.0278 |
| | | (0.0463) [9.04%] | (0.0468) | (0.0510) | (0.0513) |
| *Frequency of eating out* (Base: Seldom) | | | | | |
| • Often | | **0.284**\*\*\* | **0.165**\*\*\* | **0.171**\*\*\* | **0.170**\*\*\* |
| | | (0.0494) [32.84%] | (0.0516) [17.94%] | (0.0571) [18.65%] | (0.0572) [18.53%] |
| • Normal | | **0.172**\*\*\* | **0.0805**\* | **0.100**\*\* | **0.108**\*\* |
| | | (0.0407) [18.77%] | (0.0422) [8.38%] | (0.0461) [10.52%] | (0.0463) [11.40%] |
| *Age* (Base: 65 or older) | | | | | |
| • 18 to 24 | | | **0.558**\*\*\* | **0.456**\*\*\* | **0.471**\*\*\* |
| | | | (0.0647) [74.72%] | (0.101) [57.78%] | (0.101) [60.16%] |
| • 25 to 34 | | | **0.380**\*\*\* | **0.238**\*\* | **0.248**\*\* |
| | | | (0.0651) [46.23%] | (0.0960) [26.87%] | (0.0962) [28.15%] |
| • 35 to 44 | | | **0.258**\*\*\* | 0.0871 | 0.0991 |
| | | | (0.0651) [29.43%] | (0.0935) | (0.0940) |
| • 45 to 54 | | | **0.112**\* | -0.0504 | -0.0451 |
| | | | (0.0664) [11.85%] | (0.0936) | (0.0939) |
| • 55 to 64 | | | -0.0196 | -0.129 | -0.125 |
| | | | (0.0691) | (0.0874) | (0.0875) |
| New Zealand | | | -0.0374 | -0.0498 | -0.0371 |
| | | | (0.0442) | (0.0479) | (0.0481) |
| Female | | | -0.0236 | 0.00214 | 0.00764 |
| | | | (0.0357) | (0.0399) | (0.0403) |
| *Relationship status* (Base: Single) | | | | | |
| • In a Relationship | | | | -0.0699 | -0.0730 |
| | | | | (0.0530) | (0.0531) |
| • Married | | | | 0.0207 | 0.0186 |
| | | | | (0.0523) | (0.0524) |
| • Child | | | | 0.0363 | 0.0337 |
| | | | | (0.0494) | (0.0495) |
| *Education status* (Base: Did not complete high school) | | | | | |
| • High School | | | | 0.123 | 0.135 |
| | | | | (0.0839) | (0.0840) |
| • Certificate / Diploma | | | | 0.132 | **0.141**\* |
| | | | | (0.0822) | (0.0824) [15.14%] |
| • Bachelor / Post-graduate | | | | **0.188**\*\* | **0.195**\*\* |
| | | | | (0.0859) [20.68%] | (0.0858) [21.53%] |
| *Employment status* (Base: Full-time employed) | | | | | |
| • Part-time employed | | | | -0.0251 | -0.0379 |

*(Continued)*

**Table 5.** (Continued)

| Variable | Model 1 | Model 2 | Model 3 | Model 4 | Model 5 |
|---|---|---|---|---|---|
| | | | | (0.0527) | (0.0528) |
| • Retired | | | | **-0.187**** | **-0.185**** |
| | | | | (0.0859) [-17.06%] | (0.0908) [-16.89%] |
| • No employment | | | | -0.0465 | -0.0588 |
| | | | | (0.0568) | (0.0570) |
| *Household income* (Base: Less than $40K) | | | | | |
| • $40K - $60K | | | | 0.0834 | **0.0931*** |
| | | | | (0.0562) | (0.0564) [9.76%] |
| • $61K - $85K | | | | 0.0162 | 0.0213 |
| | | | | (0.0656) | (0.0656) |
| • $86K - $100K | | | | -0.0488 | -0.0322 |
| | | | | (0.0699) | (0.0703) |
| • $100K or more | | | | -0.0741 | -0.0780 |
| | | | | (0.0671) | (0.0673) |
| Female*Healthy Eating Goals | | | | | -0.0600 |
| | | | | | (0.0404) |
| Female*Food Economizing Goals | | | | | -0.0557 |
| | | | | | (0.0377) |
| Female*Food Hedonism Goals | | | | | **0.124**** |
| | | | | | (0.0386) [13.20%] |
| _Cons | 1.279*** | 1.067*** | 0.929*** | 0.948*** | 0.932*** |
| | (0.0176) | (0.0379) | (0.0575) | (0.126) | (0.126) |
| *N* | 1001 | 1001 | 996 | 885 | 885 |

Note: Standard errors in parentheses.

* $p < 0.10$,

** $p < 0.05$,

*** $p < 0.01$. For ease of interpretation, percentage values are included in brackets for the statistically significant estimates. For the frequency of eating out variables often represents eating out 3 times a week or more; normal stands for eating out once or twice a week; seldom corresponds to once a month or only on special occasions. Among the employment status variables 'No employment' includes those who are unemployed as well as students, those not able to work, retired people and homemakers.

goals and the food economizing variable. In summary, healthy eating goals has the largest positive effect on the WTPP. On the other hand, food economizing and food hedonism both decrease consumers' WTPP for healthy beverages.

The coefficients of the censored Poisson regression can be interpreted as the difference in the logs of the expected amount of 'premium' to be paid for a healthy beverage. For example, a one unit increase in the factor scores of healthy eating goals means the log of the expected premium would increase by 0.170 to 0.205 units, while holding other variables constant. This number can also be interpreted as a percentage by exponentiating the coefficient and then subtracting one from it. The coefficients for the healthy eating goals variable would correspond to a 19–23% increase in the willingness to pay a premium. However, the coefficients for the food economizing and food hedonism variables would represent reductions of 7–10% and 4–6% respectively.

Most of the control variables in our regression analysis have expected signs and statistical significance. Consumers spending $21-$35 when eating out are willing to pay more for healthy beverages. Our explanation for this as at an expenditure amount of $36 +, the purchase

expands beyond a standard meal and drink and but most likely includes wine or desert, in which case healthy eating goals are superseded by enjoyment of the dining experience.

WTPP for healthy beverages increases with eating out frequency. Compared to those who seldom eat out (once a month or on special occasions), consumers who eat out regularly/normal (1–2 a week) and often (3 or more times a week) are more willing to pay a premium for a healthy beverage. The premium price they are willing to pay also increases with their eating out frequency; those who eat out more often are willing to pay a larger premium. Similarly, WTPP for healthy beverages decreases with age, with the youngest cohort (aged 18–24) willing to pay the largest premium for healthy beverages. We also observe a positive association with higher education and our dependent variable, but only for respondents with a Bachelor's degree or higher. Other control variables do not show any statistical significance, with one exception–the coefficient for those whose employment status is 'retired' is consistently negative and significant. The results for the income variables are also worth mentioning. Contrary to the previous findings in the literature, our results suggest almost no association between income and WTPP, except for a marginally significant effect for the $40K - $60K group, compared to those who make less than $40K. However, this result is in accordance with the findings for the consumers' spending when eating out. Together, these results might suggest an inverse-U shaped relationship between expenditure when eating out, income and WTPP for healthy beverages.

Table 5 provides interesting gender effects. First, our results suggest no direct gender effect on WTPP. In other terms, keeping everything else constant, we observe no differences between males and females in their willingness to pay a premium for healthy beverages. More interestingly, Model 5 shows that gender has no moderating effect on the relationship between health eating goals and WTPP, or food economizing goals and WTPP. However, looking at the interaction of gender and food hedonism goals suggests some nuances, capturing gender heterogeneity in food hedonism. The relationship between food hedonism and WTPP is significant and negative for men, but switches sign when it comes to women; relative to men, women with a similar measure of food hedonism are more willing to pay a premium for healthy beverages. This can be explained by different perceptions among women as to the association between healthy products and hedonistic consumption.

Lastly, we have performed power analysis for our smallest sample size of 885 observations (Models 4 and 5). The post-hoc power test for N = 885 is 1, suggesting sufficiency of the number of observations for our analysis.

## Hypotheses tests

Results of the hypotheses test are summarized in Table 6. Our results found support for three out of the six hypotheses examined in this study.

## Barriers to healthy beverage purchasing

In addition to examining the predictors of WTPP for healthy beverages and testing the study hypotheses, **RQ3** of this study sought to explain "What are the obstacles faced by consumers when buying healthy beverages in hospitality businesses?". Respondents were asked to indicate the obstacles that restricted them from ordering healthier drinks. Response categories related to price, taste, availability of product options, familiarity with products and brands, and perceptions toward the health claim credibility (Table 7). Respondents were free to list more than one barrier; therefore, the percentages within the table represent the percentage of responses received for a particular barrier category across all received responses.

Table 6. Hypotheses test results.

| Hypothesis | Supported/ Not supported | Description |
|---|---|---|
| **H₁: Women will have a significantly higher willingness to pay a premium for healthy beverage products** | Not supported | We find no direct gender effect. In other terms, after controlling for various factors, there is no difference between men and women in terms of WTPP. |
| **H₂: Older consumers will have a significantly higher willingness to pay a premium for healthy beverage products as compared to younger consumers** | Not supported | We find the opposite. Younger consumers have a higher willingness to pay a premium for healthy beverage products. |
| **H₃: Level of income will have positive relationship with an individual's willingness to pay a premium for healthy beverage products** | Not supported | After controlling for various factors, we find that the level of income has no significant association with WTPP. |
| **H₄: Consumers' motivations toward health eating will have significant positive effect on willingness to pay a price premium for healthy beverages** | Supported | Individual attitudes and goals toward healthy eating significantly increases the WTPP of the respondents, and we observe no distinction between males and females. |
| **H₅: Consumers' motivations toward food hedonism will have a significant negative effect on willingness to pay a price premium for healthy beverages.** | Supported | While not as strong and robust as the other attitude variables, we observe a negative relationship between hedonism goals and WTPP (across the aggregate data). |
| **H₆: Consumers' motivations toward food economizing will have a significant negative effect on willingness to pay a price premium for healthy beverages** | Supported | After controlling for a variety of variables, we show that individual goals toward food economizing significantly decreases the WTPP of the respondents, and we observe no distinction between males and females. |

Results revealed 431 responses to the 'barrier' of 'healthy drinks cost more'. In addition, 368 responses identified a 'limited range of healthy beverage product options'. There were 227 responses to the category of 'healthy drinks do not taste as good'. Interestingly, there were 314 responses from the sample regarding 'not convinced that healthy drinks are truly healthy', highlighting some consumer skepticism toward marketing claims of health and wellbeing regarding beverage products.

## Discussion

This empirical study of hospitality consumers' attitudes towards healthy beverages and their willingness to pay a premium addresses four overarching research questions: **RQ1**: Which beverages that are sold in hospitality businesses do consumers consider to be 'healthy'? **RQ2**: What criteria do consumers use to determine whether a beverage is considered to be 'healthy'? **RQ3**: What are the obstacles faced by consumers when buying healthy beverages in hospitality

Table 7. Barriers to purchasing healthy beverages.

| Obstacle | N | % |
|---|---|---|
| Cost | 431 | 29.28 |
| Limited product options | 368 | 25.00 |
| Not convinced that healthy drinks are truly healthy | 314 | 21.33 |
| Healthy drinks do not taste as good | 227 | 15.42 |
| Not familiar with the brands and products | 132 | 8.97 |
| Other | 83 | 8.1 |

businesses? **RQ4**: What are the motivational, demographic, and behavioral variables that influence consumers' willingness to pay a price premium for healthy (vs. unhealthy) beverages?

Our descriptive analysis of over 1000 responses reveals interesting insights into what constitutes a healthy beverage from a hospitality consumer perspective. In this approach, the term 'healthy' was not defined or made explicit to respondents to openly explore what beverages are perceived as 'healthy', and the attributes that determine this assessment. Not surprisingly, water and fruit juice (including freshly squeezed juices) were the highest reported health drinks sold in hospitality establishments. Other products considered healthy include teas and smoothies; however, less than 2% of our sample identified sugar-free soft drinks as being healthy despite their minimal sugar content. Regarding the attributes that determine the healthiness of a beverage, almost half our respondents identified beverages with 'low sugar' or 'no sugar' as key criteria. In addition, 42% of respondents identified products that are 'natural' or 'free of additives and preservatives' to be healthy. Overall, there were minimal reports of products that contain 'probiotics' or are 'organic' as being healthy, despite the widespread distribution and promotion of these beverages as health products. These findings are consistent with recent reports that document demand for products positioned as 'all natural' are gaining attention from consumers [70].

RQ4 of the study sought to examine the motivational (goals), demographic, and behavioral variables that influence consumers' willingness to pay a price premium for healthy (vs unhealthy) beverages. Our results reveal that a higher frequency of eating out is positively associated with consumers' WTPP for healthy beverages. Drawing on the marketing and food literature, our study hypothesized that 1) Women will have a significantly higher willingness to pay a price premium for healthy beverage products compared to men; and 2) Older consumers will have a significantly higher willingness to pay a price premium for healthy beverage products as compared to younger consumers. Interestingly, neither of these hypotheses could be supported. While some previous studies suggest women are likely to have a higher preference for healthy products and are more likely to purchase these regularly [36, 37, 44], our findings are more consistent with others that have found no role of gender determining consumers' WTPP for healthy/organic food products [38, 39]. Moreover, in the context of organic wines some research suggests women are less inclined to pay a premium [51]. Thus, the effects of gender in determining WTPP for health products remains inconclusive, and marketers should focus more on motivational and behavioral characteristics. A similar case can be made for age groups. Our findings that younger (18–24-year-old) consumers have a higher willingness to pay a premium for healthy beverages than older consumers in contrast with previous literature [47, 48]. This suggests that younger segments of the market now have a higher awareness of diet, health, and nutrition, and again emphasizing that psychographic variables and motivations goals are far more pertinent in understanding purchase behaviors than age or gender.

Our results also found that motivational goals included healthy eating, food economizing, and food hedonism [59, 63] are significantly related to WTPP, but in different ways. Censored Poisson regression revealed that Healthy Eating Goals has a positive and significant effect on (WTPP); a one-unit increase in Healthy Eating Goals is associated with around a 20% increase in WTPP. In contrast, motivations related to consuming foods for hedonistic purposes, and motivations driven by economizing goals, are negatively related to a willingness to pay a premium. A higher level of Food Economizing Goals is associated with around a 10% decrease in WTPP. A similar negative effect is also present for Food Hedonism Goals but at a smaller magnitude; a one-unit increase in Food Hedonism is associated with around a 5% decrease in willingness to pay.

These findings present new insights on how psychographic variables influence the purchase intentions of hospitality consumers. Specifically, while previous studies have examined these

relationships in the context of food purchases and in supermarket/retail environments [see 59, 64], our findings extend the literature to present new contributions on consumer behavior when purchasing from hospitality venues. Clearly, while some consumers may be motivated to purchase healthy at the expense of affordability and taste, others are driven by the price and taste/enjoyment of the purchase. This presents implications for market segmentation and strategies in the food and hospitality sectors.

In addition to the understanding of what beverages consumers consider to be healthy and their associated attributes, RQ3 in this study sought to uncover the barriers facing consumers when purchasing healthy beverages in a hospitality context. The barriers reported by our sample to purchasing healthy beverages included 1) the cost of the beverages, 2) limited product availability; and 3) not being convinced that healthy beverages are truly healthy. Moreover, only 15% of responses indicated that healthier beverages did not taste as good, suggesting a greater acceptance by consumers for healthier and more natural tasting products without the need for added sugars or artificial sweeteners.

Our findings, at least at the sample level, suggest a growing acceptance by hospitality consumers of the taste associated with healthier beverage options. Of course, there will be a segment of the market who remain doubtful of the real healthiness of 'healthy' beverage products; as is the case in our sample. Hospitality consumers have developed greater knowledge toward health, nutrition and healthy product attributes. Hospitality firms and beverage manufacturers are now dealing with well informed consumers who will read product labels and assess ingredients in terms of sugars, additives and artificial ingredients. This suggests the healthiness of the product should be scientifically supported and validated by reputable bodies [71]. This is where front of package food labelling and health star ratings can influence consumer decision making [30].

Another reported barrier to purchasing as reported by our sample was the limited range of healthy beverage product options available in hospitality venues. This is in comparison to the mass availability and range of high calorie, sugar-sweetened beverages, or articially sweetened beverages, that remain prevalent in the hospitality industry [21]. For example, over 75% of the 50 largest restaurant chains in the US have softdrinks and other sugar sweetened drinks as the default beverage choice on their children's menu [72]. This highlights opportunities for innovation in new product development and wider distribution across the hospitality sectors.

Finally, the extant literature suggests women are more inclined to make healthy food and drink choices and to purchase more regularly [36, 37, 44]. Our study expands on the body of knowledge by examining if the relationships among motivational goals (healthy eating, economizing, hedonism) and WTPP are moderated by gender differences. Results of the interaction effects found gender to have no significant moderating effects on the relationship between health goals and WTPP, or economizing goals and WTPP. However, there is evidence of moderation between hedonism goals and WTPP, suggesting different perceptions among women as to the association between healthy products and hedonistic consumption. The relationship between food hedonism and WTPP is negative for men, but switches sign when it comes to women. Men that are high on the food hedonism scale are less likely to pay a premium for healthy beverages. Compared to men, however, women with similar food hedonism scores are willing to pay more of a premium.

## Conclusion, limitations, and future research directions

The widespread distribution of high calorie sugar-sweetened beverages in the hospitality industry continues to have negative effects on public health [19]. The hospitality industry and foodservice sector can play an important role in the development and supply of healthier

options, nudging consumers to choose healthy. Understanding the healthy beverage sector from the consumers' perspectives provides valuable insights to the hospitality industry and business operators.

Regarding implications for policy and practice, our study highlights the need for a collaborative effort among government, beverage manufacturers, and the hospitality sector in increasing the variety and supply of healthy product options. Evidence of strong consumer demand and willingness to pay should drive innovation in new product development and distribution, bringing products that are healthy, taste good, and reasonably priced to hospitality consumers. These measures will address the reported barriers to purchasing, as revealed in our study, regarding the limited availability of healthy beverage products sold in hospitality venues. In addition, consumer skepticism around 'health' claims of products should also be addressed by providing validated information about the product's attributes. Beverages should include Front of Package (FoP) nutritional labeling, such as certified Health Star Ratings, as this information is perceived to be more credible than marketing communication [30]. The strong links between consumers' healthy eating goals and WTPP suggests a need for Government-led social marketing initiatives that influence people's motivations and attitudes toward health and wellbeing. These cognitive factors are strong drivers of intention to purchase healthier products and encouraging health conscious (as opposed to hedonic) decisions.

In terms of limitations, data for this study was collected from hospitality industry consumers in Australia and New Zealand. Our insights on what consumers determine to be healthy beverages and their attributes are based on descriptive analysis of the sample responses, thus, the generalizability of these findings to the broader population needs further testing. Nevertheless, it is important to note that our sampling design was representative of the broader population for each country. We encourage future studies to collect data from different countries and examine cross-cultural/country differences among consumers' purchase behaviors. Larger samples could also be collected to allow for a multigroup comparison between respondents across various psychological and demographic factors. In addition, an in depth understanding of how consumers make judgments regarding the healthiness of a beverage, and what information they use to make these assessments, is warranted to understand consumers' knowledge of health and nutrition and its impact on consumption.

As noted in this study the definition of 'healthy' was not stipulated in to order to gain insights through respondents' self-evaluations. Future research aiming to examine objective measures of 'healthiness' could use more explicit criteria based on the product's quantifiable attributes, ingredients and nutritional profile (e.g. amount of sugar, calories, fat etc.). Despite these limitations, this study presents new contributions to the body of knowledge on consumers' attitudes, motivations, and purchase behaviors concerning healthy beverages.

## Supporting information

**S1 Appendix. Summary statistics of the sample.**
(DOCX)

**S2 Appendix. Chi-square difference tests for age and gender distribution.**
(DOCX)

**S3 Appendix. Questionnaire.**
(DOCX)

**S1 Data.**
(DTA)

## Acknowledgments

The authors would like to thank the University of South Australia, Le Cordon Bleu Australia, and Organic and Raw Trading Company Pty Ltd for supporting this project. We would also like to thank Chameleon Customer Contact for their assistance with the data collection.

## Author Contributions

**Conceptualization:** Rob Hallak, Ilke Onur.

**Formal analysis:** Rob Hallak, Ilke Onur, Craig Lee.

**Funding acquisition:** Rob Hallak.

**Investigation:** Rob Hallak.

**Methodology:** Rob Hallak, Ilke Onur, Craig Lee.

**Project administration:** Rob Hallak, Craig Lee.

**Validation:** Rob Hallak, Ilke Onur.

**Writing – original draft:** Rob Hallak, Ilke Onur, Craig Lee.

**Writing – review & editing:** Rob Hallak, Ilke Onur, Craig Lee.

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
