## [Decision Letter · Decision Letter 0]

28 Sep 2021

PONE-D-21-21464Consumer demand for healthy beverages in the hospitality industry: Examining willingness to pay and barriers to purchasePLOS ONE

Dear Dr. Hallak,

Thank you for submitting your manuscript to PLOS ONE. After careful consideration, we feel that it has merit but does not fully meet PLOS ONE’s publication criteria as it currently stands. Therefore, we invite you to submit a revised version of the manuscript that addresses the points raised during the review process.

We look forward to receiving your revised manuscript.

Kind regards,

Maya K. Vadiveloo

Academic Editor

PLOS ONE

2. Please include additional information regarding the survey or questionnaire used in the study and ensure that you have provided sufficient details that others could replicate the analyses. For instance, if you developed a questionnaire as part of this study and it is not under a copyright more restrictive than CC-BY, please include a copy, in both the original language and English, as Supporting Information. Furthermore, please provide additional information regarding how the questionnaire was distributed online.

3. During our internal checks, the in-house editorial staff noted that you conducted research or obtained samples in another country. Please check the relevant national regulations and laws applying to foreign researchers and state whether you obtained the required permits and approvals. Please address this in your ethics statement in both the manuscript and submission information.

Additional Editor Comments (if provided):

Thank you for your submission. Both reviewers commend the authors for the writing style and significance of the research question. However, a number of major concerns were raised that must be addressed before the manuscript can be considered for publication.

Reviewers' comments:

Reviewer's Responses to Questions

**Comments to the Author**

1. Is the manuscript technically sound, and do the data support the conclusions?

Reviewer #1: Partly

Reviewer #2: Yes

2. Has the statistical analysis been performed appropriately and rigorously? 

Reviewer #1: No

Reviewer #2: Yes

3. Have the authors made all data underlying the findings in their manuscript fully available?

Reviewer #1: Yes

Reviewer #2: No

4. Is the manuscript presented in an intelligible fashion and written in standard English?

Reviewer #1: Yes

Reviewer #2: Yes

5. Review Comments to the Author

Reviewer #1: The authors offer very interesting insight into consumers’ conceptualisation of healthy beverages and into their willingness to pay a little extra compared to unhealthier drink options. The writing style is clear and elegant and some findings are clearly interesting; however, this manuscript does not present a body of evidence that clearly advances the literature. The six hypotheses asks questions that the authors already answered in the Introduction; there are also some issues in regards to some methodological choices and the discussion does not offer new significant insight. I will now explore these issue in more detail below.

Overall, the paper is very well-written but it is repetitive in quite a few places, for example (a)

lines 60-62 on p.3, (b) early on p.6 and (c) lines 337-341 on p.16. In other places, the manuscript presents some relatively or uninformative elements, for example (d) lines 106-111 on p.5, (e) lines 482-489 on p.29, and (f) lines 569-583 on p.33. Relatedly, the Hypotheses Tests presented at Table 6 should be integrated within the text in an effort to streamline the manuscript.

As mentioned above, all 6 hypotheses do not seem to explore original questions as the literature the authors discuss already addressed these and thus no incremental value can be easily ascertained. As a related example, on p.8 (line 176) the authors state that “there are myriad of factors that may contribute…” but then to do not explore what these factors are; it is therefore arduous to appreciate what novel questions and knowledge are offered in the manuscript.

The Method is a well-presented section, although a few choices should be further justifies. For example, it is not entirely clear why only consumers who eat out are included in this study; after all, selection of healthy (vs. unhealthier) drinks can be made in supermarkets, too – and, overall, it is likely to represent a greater issue in terms of public’s well-being.

There is no justification (e.g., power analysis) for the sample size tested in this study.

A rationale and more in-depth justification should be offered in terms of the selection of items for the three predictors; for example, it is unclear whether the items were selected from previously utilised (and reliable) measures or not. Although the authors offer some reassurance through factor analysis, other types of validity are not discussed nor addressed.

There are further issues with the selection of the items. The authors state that their strategy was not to mention what a healthy drink is in order to avoid constraining participants’ perceptions. However, items in the Healthy Eating Goals measure indicate specific elements (e.g., containing vitamins and minerals, being good for appearance, being nutritious) that can prime participants in their answers. Other elements that should be addressed is whether foods that are good for the appearance (item for the Healthy Eating Goals measure) are necessarily healthy. It is also somewhat surprising that the criterion variable (WTPP) is about drinks but the predictor measures regard food.

Further explanations could be offered about the WTPP, too (e.g., the choice of $5 as the comparison point). It is also not entirely clear how the choice data (i.e., the unhealthy $5 vs. the alternative) has been combined with the frequency data (“never”, “occasionally”, or “most of the time”) in order to categorise participants. Is the switch point determine when a participant switches from “never” to “occasionally” or from the latter to “most of the times”? The mention of the “maximum” price that participants were willing to pay (p.13) does not elucidate this issue.

In terms of the Control variables, it is unclear why Country has been included and why would the authors expect differences between AUS and NZ. The same question arises when employment and marital status and having children are included in the analyses, too.

Whilst censored regression seems a reasonable choice, it is not clear why the authors have not chosen a more straightforward alternative such as ordinal regression, which could be appropriate for this data set, too.

It is also not accessible to the reader what the purpose of the model and the equations presented on pp.14-16. In their current form, these considerations seem to refer to general formulations of the Poisson regression analysis that therefore offer no information that is specific to this manuscript and its analysis. Are the authors justifying the selection of a particular log likelihood function that has been amended on the basis of the purpose of this specific analysis? If the equations and considerations are general and bear no identifying information for the specifics of the present analysis, they could be omitted.

The data stemming from the open questions, due to issues of non-independence, are not analysed inferentially and thus conclusions drawn on descriptive analyses should be offered more tentatively, which weakens the manuscript. Further details would benefit the manuscript as well. For example, on Table 2 the total count is 1,373 responses and thus a breakdown of whether all participants provided at least a response and how many offered multiple responses, and how many of them could be presented. Table 3 offers half as many total responses (776) so it is clear that the rate of non-responding was higher for this item. The same comment applies to the data summarised on Table 7.

In hierarchical regression models, usually the predictors of interest (in this case food hedonism, economising and healthy eating goals) are entered in the last step(s) in order to explore whether they can explain any residual variance that is not accounted for by known or control predictors. Why is it the other way around here?

The authors could also offer further explanations of findings. For example, the authors could explore why Food Hedonism becomes non-significant in Models 3 and 4 only (at the moment, only a general comment about the strength of the effect of WTPP is offered; p.25, lines 423-424). These considerations could be offered by exploring the associations with the variables entered at different stages.

It is also unclear why, in Model 5, the authors explored the interaction effects only for Gender and not for control variables and demographics. The explanations regarding the observed effects warrant further elaboration, too – as at the moment there is a only a mention that the relationship between hedonism and WTPP “changes for women” and it is due to “different perceptions among women” (p.26, line 455-457).

Related to the above, other effects should be discussed further. For example, although spending when eating out matters, it is interesting to note that this is true only for the $21-$35 vs. $20 comparison, but not for the $36 vs. $20 comparison, which seems counter-intuitive. The same applies for household income, where only the $40-$60k v < $40k comparison is significant, but none of the other income groups.

The Discussion seems to focus much on summarising the outcomes rather than elaborating on additional insight (e.g., p.31). As mentioned above, some conclusions are based only on descriptive statistics and thus should be formulated more tentatively. Other arguments (e.g., about potential differences between AUS and NZ, lines 584-588 on p.33) are only hinted at, too. Relatedly, some arguments offered in regards to limitations should be explored further, for example in terms of sample size and the use of more explicit criteria for categorising healthy beverages, which has been previously mentioned as a strategic choice of the study.

Minor comments

p.8, line 185: Define what hedonistic approach towards food is as soon as you introduce this construct

p.14, line 300: It is not clear why the authors mention that the count data “are likely” to be skewed; after all, the authors have these data and thus should be reporting on them more specifically.

p.17: The authors may offer additional detail about coding of answers, for example whether multiple and independent assessors were required to ensure consistency in terms of responses to open questions.

On the regression table, I wonder whether the authors could enter information about percentages (as explained in lines 428-436) so that the reader can more easily appraise the size of the effect of each predictor.

Reviewer #2: Review for Consumer demand for healthy beverages in the hospitality industry: Examining willingness to pay and barriers to purchase

This is an interesting and useful study. The paper is well written and results are clearly presented, but the discussion conclusions sections lack some clarity. This study should be of interest to the readers of this journal and I support the publication of the paper with some revisions to address some of the major comments below.

Major comments

First, the study will be most useful for NZ and Au policy makers and hospitality industries and less applicable to other parts of the world, which is recognized by the authors as a limitation. How do you believe your study areas are different from other areas? What features may be unique? If you want readers to understand how generalizable the results are or which ones are and which are not, more discussion would be helpful.

Second, although their literature review supports a positive correlation of income with WTPP, hence their H3, the authors spend no time providing any intuition as to why they found no support in their data for H3. I think it is important to dedicate some discussion space to this result. Similarly, H1 and H2 also need more discussion to understand why the data in this study contradicts prior research. On lines 534 and 535 there is an explanation about the gender effects regarding the different perceptions among women but it is left to the reader to guess what those different perceptions are in fact. This explanation is repetitive (lines 455, 456, 457) and the discussion offers no more insight than results section.

Third, it would be good to see the policy outcomes spelled out a bit more clearly and broadly (based on the three hypotheses for which they did find support).

Fourth, there are other papers that aren’t mentioned in the paper that the authors might consider.

Ali, Tabassum, and Jabir Ali. “Factors Affecting the Consumers’ Willingness to Pay for Health and Wellness Food Products.” Journal of Agriculture and Food Research 2 (2020): 100076. https://doi.org/10.1016/j.jafr.2020.100076.

A. Krystallis, G. Chryssohoidis

Consumers' willingness to pay for organic food: factors that affect it and variation per organic product type Br. Food J., 107 (5) (2005), pp. 320-343

S. Gao Effects of additional quality attributes on consumer willingness to pay for food labels

Ann. Phys. (2007), pp. 1-133

M.L. Loureiro, J.J. McCluskey, R.C. Mittelhammer Will consumers pay a premium for eco-labeled apples? J. Consum. Aff., 36 (2) (2002), pp. 203-219

Ogbeide, Osadebamwen. (2015). Consumer Willingness to Pay a Premium for the Health Benefits of Organic Wine. Mayfair Journal of Agribusiness Management. 1. 1-23.

Minor editing suggestions – typos and grammatical errors

Line 32 – finds instead of find

Line 67 – there are also growing concerns

Line 144 “…established than younger people’s”

Line 265 “ through” is unnecessary

Line 354 – 66% of respondents

Line 356 – almost 79% of respondents spent

Line 357 – consistent with Australian

Line 493 – which have established health benefits

Line 549 – sugar sweetened beverages

Line 550 - that remain prevalent

Line 552 – sugar sweetened drinks

Line 575 – food service sector

6. PLOS authors have the option to publish the peer review history of their article (what does this mean?). If published, this will include your full peer review and any attached files.

Reviewer #1: No

Reviewer #2: No

---

## [Author Response · Author response to Decision Letter 0]

16 Mar 2022

Please see attached our revised submission of an original Research Article titled “CONSUMER DEMAND FOR HEALTHY BEVERAGES IN THE HOSPITALITY INDUSTRY: EXAMINING WILLINGNESS TO PAY A PREMIUM, AND BARRIERS TO PURCHASE” for consideration in PLOS ONE. Thank you for the opportunity to revise our manuscript and for the extensive and thorough feedback received from the reviewers. In total there were 28 points for revisions, we have addressed all comments and made substantial improvements to our manuscript. The quality, rigor, and contribution of the paper has improved because of the review process.

---

## [Decision Letter · Decision Letter 1]

14 Apr 2022

CONSUMER DEMAND FOR HEALTHY BEVERAGES IN THE HOSPITALITY INDUSTRY: EXAMINING WILLINGNESS TO PAY A PREMIUM, AND BARRIERS TO PURCHASE

PONE-D-21-21464R1

Dear Dr. Hallak,

We’re pleased to inform you that your manuscript has been judged scientifically suitable for publication and will be formally accepted for publication once it meets all outstanding technical requirements.

Kind regards,

Maya K. Vadiveloo

Academic Editor

PLOS ONE

Additional Editor Comments (optional):

Reviewers' comments:

Reviewer's Responses to Questions

**Comments to the Author**

1. If the authors have adequately addressed your comments raised in a previous round of review and you feel that this manuscript is now acceptable for publication, you may indicate that here to bypass the “Comments to the Author” section, enter your conflict of interest statement in the “Confidential to Editor” section, and submit your "Accept" recommendation.

Reviewer #1: All comments have been addressed

Reviewer #2: All comments have been addressed

2. Is the manuscript technically sound, and do the data support the conclusions?

Reviewer #1: Yes

Reviewer #2: Yes

3. Has the statistical analysis been performed appropriately and rigorously? 

Reviewer #1: Yes

Reviewer #2: Yes

4. Have the authors made all data underlying the findings in their manuscript fully available?

Reviewer #1: Yes

Reviewer #2: Yes

5. Is the manuscript presented in an intelligible fashion and written in standard English?

Reviewer #1: Yes

Reviewer #2: Yes

6. Review Comments to the Author

Reviewer #1: The authors comprehensively addressed all of my comments and have considerably improved the quality of the manuscript.

The only concern that should be addressed further regards power analysis. Observed power (post-hoc power; lines 552-554) directly relates to p-values and thus it is uninformative (e.g., Althouse, 2021; Hoenig & Heisey, 2001). More specifically, Hoenig and Heisey (2001; p.2) conclude that “Observed power can never fulfill the goals of its advocates because the observed significance level of a test also determines the observed power; for any test, the observed power is a 1:1 function of the P value”.

Relatedly, in their Response 1.7 the authors report that they considerably over-recruited as they indicated that N = 248 would have sufficed to achieve the desired level of power. As a result, some effects that are very small (e.g., a coefficient of .0382) come out as significant.

I thus advise the authors to add a cautionary note about the above in the manuscript and to apply caution when interpreting very small coefficients.

• Althouse A. (2021). Post Hoc Power: Not Empowering, Just Misleading. The Journal of Surgical Research, 259, 3-6.

• Hoenig JM & Heisey DM. (2001). The abuse of power: the pervasive fallacy of power calculations for data analysis. The American Statistician, 55, 19-24.

Reviewer #2: I am satisfied with the authors' answers to my revision requests and have no further comments.

7. PLOS authors have the option to publish the peer review history of their article (what does this mean?). If published, this will include your full peer review and any attached files.

Reviewer #1: No

Reviewer #2: No

---

## [Editor Report · Acceptance letter]

22 Apr 2022

PONE-D-21-21464R1 

Consumer demand for healthy beverages in the hospitality industry: Examining willingness to pay a premium, and barriers to purchase 

Dear Dr. Hallak:

I'm pleased to inform you that your manuscript has been deemed suitable for publication in PLOS ONE. Congratulations! Your manuscript is now with our production department. 

Kind regards, 

on behalf of

Dr. Maya K. Vadiveloo 

Academic Editor

PLOS ONE